# A Copula approach for hyperparameter transfer learning

## Abstract

Bayesian optimization (BO) is a popular methodology to tune the hyperparameters of expensive black-box functions. Despite its success, standard BO focuses on a single task at a time and is not designed to leverage information from related functions, such as tuning performance objectives of the same algorithm across multiple datasets. In this work, we introduce a novel approach to achieve transfer learning across different *datasets* as well as different *objectives*. The main idea is to regress the mapping from hyperparameter to objective quantiles with a semi-parametric Gaussian Copula distribution, which provides robustness against different scales or outliers that can occur in different tasks. We introduce two methods to leverage this estimation: a Thompson sampling strategy as well as a Gaussian Copula process using such quantile estimate as a prior. We show that these strategies can combine the estimation of multiple objectives such as runtime and accuracy, steering the optimization toward cheaper hyperparameters for the same level of accuracy. Experiments on an extensive set of hyperparameter tuning tasks demonstrate significant improvements over state-of-the-art methods.

## 1 Introduction

Tuning complex machine learning models such as deep neural networks can be a daunting task. Object detection or language understanding models often rely on deep neural networks with many tunable hyperparameters, and automatic hyperparameter optimization (HPO) techniques such as Bayesian optimization (BO) are critical to find the good hyperparameters in short time. BO addresses the black-box optimization problem by placing a probabilistic model on the function to minimize (e.g., the mapping of neural network hyperparameters to a validation loss), and determine which hyperparameters to evaluate next by trading off exploration and exploitation through an acquisition function. While traditional BO focuses on each problem in isolation, recent years have seen a surge of interest in *transfer learning* for HPO. The key idea is to exploit evaluations from previous, related *tasks* (e.g., the same neural network tuned on multiple datasets) to further speed up the hyperparameter search.

A central challenge of hyperparameter transfer learning is that different tasks typically have different scales, varying noise levels, and possibly contain outliers, making it hard to learn a joint model. In this work, we show how a semi-parametric Gaussian Copula can be leveraged to learn a joint prior across datasets in such a way that scale issues vanish. We then demonstrate how such prior estimate can be used to transfer information across tasks and objectives. We propose two HPO strategies: a Copula Thompson Sampling and a Gaussian Copula Process. We show that these approaches can jointly model several objectives with potentially different scales, such as validation error and compute time, without requiring processing. We demonstrate significant speed-ups over a number of baselines in extensive experiments.

The paper is organized as follows. Section 2 reviews related work on transfer learning for HPO. Section 3 introduces Copula regression, the building block for the HPO strategies we propose in Section 4. Specifically, we show how Copula regression can be applied to design two HPO strategies, one based on Thompson sampling and an alternative GP-based approach. Experimental results are given in Section 5 where we evaluate both approaches against state-of-the-art methods on three algorithms. Finally, Section 6 outlines conclusions and further developments.

## 2    RELATED WORK

A variety of methods have been developed to induce transfer learning in HPO. The most common approach is to model tasks jointly or via a conditional independence structure, which has been been explored through multi-output GPs (Swersky et al., 2013), weighted combination of GPs (Schilling et al., 2016; Wistuba et al., 2018; Feurer et al., 2018), and neural networks, either fully Bayesian (Springenberg et al., 2016) or hybrid (Snoek et al., 2015; Perrone et al., 2018; Law et al., 2018). A different line of research has focused on the setting where tasks come over time as a sequence and models need to be updated online as new problems accrue. A way to achieve this is to fit a sequence of surrogate models to the residuals relative to predictions of the previously fitted model (Golovin et al., 2017; Poloczek et al., 2016). Specifically, the GP over the new task is centered on the predictive mean of the previously learned GP. Finally, rather than fitting a surrogate model to all past data, some transfer can be achieved by warm-starting BO with the solutions to the previous BO problems (Feurer et al., 2015; Wistuba et al., 2015b).

A key challenge for joint models is that different black-boxes can exhibit heterogeneous scale and noise levels (Bardenet et al., 2013; Feurer et al., 2018). To address this, some methods have instead focused on search-space level, aiming to prune it to focus on regions of the hyperparameter space where good configurations are likely to lie. An example is Wistuba et al. (2015a), where related tasks are used to learn a promising search space during HPO, defining task similarity in terms of the distance of the respective data set meta-features. A more recent alternative that does not require meta-features was introduced in Perrone et al. (2019), where a restricted search space in the form of a low-volume hyper-rectangle or hyper-ellipsoid is learned from the optimal hyperparameters of related tasks. Rank estimation can be used to alleviate scales issues however the difficulty of feeding back rank information to GP leads to restricting assumptions, for instance (Bardenet et al., 2013) does not model the rank estimation uncertainty while (Feurer et al., 2018) uses independent GPs removing the adaptivity of the GP to the current task. Gaussian Copula Process (GCP) (Wilson & Ghahramani, 2010) can also be used to alleviate scale issues on a single task at the extra cost of estimating the CDF of the data. Using GCP for HPO was proposed in Anderson et al. (2017) to handle potentially non-Gaussian data, albeit only considering non-parametric homoskedastic priors for the single-task and single objective case.

## 3    GAUSSIAN COPULA REGRESSION

For each task denote with $f^j : \mathbb{R}^p \to \mathbb{R}$ the error function one wishes to minimize, and with $\mathcal{D} = \{x_i, y_i\}_{i=1}^N$ the evaluations available for an arbitrary task. Given the evaluations on $M$ tasks $\mathcal{D}^M = \bigcup_{1 \le j \le M} \{x_i^j, y_i^j\}_{i=1}^{N_j}$, we are interested in speeding up the optimization of an arbitrary new task $f$, namely in finding $\arg\min_{x \in \mathbb{R}^p} f(x)$ in the least number of evaluations. One possible approach to speed-up the optimization of $f$ is to build a surrogate model $\hat{f}(x)$. While using a Gaussian process is possible, scaling such an approach to the large number of evaluations available in a transfer learning setting is challenging. Instead, we propose fitting a parametric estimate of $\hat{f}_\theta(x)$ distribution which can be later used in HPO strategies as a prior of a Gaussian Copula Process. A key requirement here is to learn a joint model, e.g., we would like to find $\theta$ which fits well on all observed tasks $f^j$. We show how this can be achieved with a semi-parametric Gaussian Copula in two steps: first we map all evaluations to quantiles with the empirical CDF, and then we fit a parametric Gaussian distribution on quantiles mapped through the Gaussian inverse CDF.

First, observe that as every $y_i$ comes from the same distribution for a given task, the probability integral transform results in $u_i = F(y_i)$, where $F$ is the cumulative distribution function of $y$. We then model the CDF of $(u_1, \ldots, u_N)$ with a Gaussian Copula:

$$C(u_1, \ldots, u_N) = \phi_{\mu, \Sigma}(\Phi^{-1}(F(y_1)), \ldots, \Phi^{-1}(F(y_N))),$$

where $\Phi$ is the standard normal CDF and $\phi_{\mu, \Sigma}$ is the CDF of a normal distribution parametrized by $\mu$ and $\Sigma$. Assuming $F$ to be invertible, we define the change of variable $z = \Phi^{-1} \circ F(y) = \psi(y)$ and $g = \psi \circ f$.[1] We regress the marginal distribution of $P(z|x)$ with a Gaussian distribution whose

---

[1]Note that if $z$ is regressed perfectly, then finding the minimum of $f$ is solved as a parameter $x$ minimizing $\psi(f(x))$ also minimizes $f(x)$ since $\psi$ is monotonically increasing.

mean and variance are two deterministic parametric functions given by

$$P(z|x) \sim \mathcal{N}(\mu_\theta(x), \sigma_\theta(x)) = \mathcal{N}(w_\mu^T h_{w_h}(x) + b_\mu, \Psi(w_\sigma^T h_{w_h}(x) + b_\sigma)),$$

where $h_{w_h}(x) \in \mathbb{R}^d$ is the output of a multi-layer perceptron (MLP) where $w_h, w_\mu \in \mathbb{R}^d, b_\mu \in \mathbb{R}, w_\sigma \in \mathbb{R}^d, b_\sigma \in \mathbb{R}$ are projection parameters and $\Psi(t) = \log(1 + \exp t)$ is an activation mapping to positive numbers. The parameters $\theta = \{w_h, w_\mu, b_\mu, w_\sigma, b_\sigma\}$ together with the parameters in MLP are learned by minimizing the Gaussian negative log-likelihood on the available evaluations $\mathcal{D}^\mathcal{M} = \bigcup_{1 \le j \le M} \{x_i^j, z_i^j\}_{i=1}^{N_j}$, e.g., by minimizing

$$\sum_{(x,z) \in \mathcal{D}^\mathcal{M}} \frac{1}{2} \log 2\pi\sigma(x)^2 + \frac{1}{2} \left( \frac{z - \mu(x)}{\sigma(x)} \right)^2 + \psi'(\psi^{-1}(z)), \tag{1}$$

with SGD. Here, the term $\psi'(\psi^{-1}(z))$ accounts for the change of variable $z = \psi(y)$. Due to the term $\psi'(\psi^{-1}(z))$, errors committed where the quantile function changes rapidly have larger gradient than when the quantile function is flat. Note that while we weight evaluations of each tasks equally, one may alternatively normalize gradient contributions per number of task evaluations.

The transformation $\psi$ requires $F$, which needs to be estimated. Rather than using a parametric or density estimation approach, we use the empirical CDF $\tilde{F}(t) = \frac{1}{N} \sum_{i=1}^{N} \mathbb{1}_{y_i \le t}$. While this estimator has the advantage of being non-parametric, it leads to infinite value when evaluating $\psi$ at the minimum of maximum of $y$. To avoid this issue, we use the Winsorized cut-off estimator

$$F(t) \approx \begin{cases} \delta_N & \text{if } \tilde{F}(t) < \delta_N \\ \tilde{F}(t) & \text{if } \delta_N \le \tilde{F}(t) \le 1 - \delta_N \\ 1 - \delta_N & \text{if } \tilde{F}(t) > 1 - \delta_N \end{cases}$$

where $N$ is the number of observations of $y$ and choosing $\delta_N = \frac{1}{4N^{1/4}\sqrt{\pi \log N}}$ strikes a bias-variance trade-off (Liu et al., 2009). This approach is semi-parametric in that the CDF is estimated with a non-parametric estimator and the Gaussian Copula is estimated with a parametric approach.

The benefit of using a non-parametric estimator for the CDF is that it allows us to map the observations of each task to comparable distributions as $z^j \sim \mathcal{N}(0, 1)$ for all tasks $j$. This is critical to allow the joint learning of the parametric estimates $\mu_\theta$ and $\sigma_\theta$, which share their parameter $\theta$ across all tasks. As our experiments will show, one can regress a parametric estimate that has a standard error lower than 1. This means that information can be leveraged from the evaluations obtained on related tasks, whereas a trivial predictor for $z$ would predict 0 and yield a standard error of 1. In the next section we show how this estimator can be leveraged to design two novel HPO strategies.

## 4 COPULA BASED HPO

### 4.1 COPULA THOMPSON SAMPLING

Given the predictive distribution $P(z|x) \sim \mathcal{N}(\mu_\theta(x), \sigma_\theta(x))$, it is straightforward to derive a Thompson sampling strategy (Thompson, 1933) exploiting knowledge from previous tasks. Given $N$ candidate hyperparameter configurations $x_1, \dots, x_N$, we sample from each predictive distribution $\tilde{z}_i \sim \mathcal{N}(\mu_\theta(x_i), \sigma_\theta(x_i))$ and then evaluate $f(x_i)$ where $i = \arg \min_i \tilde{z}_i$. Pseudo-code is given in the appendix.

While this approach can re-use information from previous tasks, it does not exploit the evaluations from the current task as each draw is independent of the observed evaluations. This can become an issue if the new black-box significantly differs from previous tasks. We now show that Gaussian Copula regression can be combined with a GP to both learn from previous tasks while adapting to the current task.

### 4.2 GAUSSIAN COPULA PROCESS

Instead of modeling observations with a GP, we model them as a Gaussian Copula Process (GCP) (Wilson & Ghahramani, 2010). Observations are mapped through the bijection $\psi = \Phi^{-1} \circ F$, where

we recall that $\Phi$ is the standard normal CDF and that $F$ is the CDF of $y$. As $\psi$ is monotonically increasing and mapping into the line, we can alternatively view this modeling as a warped GP (Snelson et al., 2004) with a non-parametric warping. One advantage of this transformation is that $z = \psi(y)$ follows a normal distribution, which may not be the case for $y = f(x)$. In the specific case of HPO, $y$ may represent accuracy scores in $[0, 1]$ of a classifier where a Gaussian cannot be used. Furthermore, we can use the information gained on other tasks with $\mu_\theta$ and $\sigma_\theta$ by using them as prior mean and variance. To do so, the following residual is modeled with a GP:

$$r(x) = \frac{g(x) - \mu_\theta(x)}{\sigma_\theta(x)}$$
$$\sim \text{GP}(m(x), k(x, x')),$$

where $g = \psi \circ f$. We use a Matérn-5/2 covariance kernel and automatic relevance determination hyperparameters, optimized by type II maximum likelihood to determine GP hyperparameters (Rasmussen & Williams, 2006). Fitting the GP gives the predictive distribution of the residual surrogate

$$\hat{r}(x) \sim \mathcal{N}(\mu_r(x), \sigma_r(x)).$$

Because $\mu_\theta$ and $\sigma_\theta$ are deterministic functions, the predictive distribution of the surrogate $\hat{g}$ is then given by

$$\hat{g}(x) = \hat{r}(x)\sigma_\theta(x) + \mu_\theta(x)$$
$$\sim \mathcal{N}(\mu_g(x), \sigma_g(x))$$
$$\sim \mathcal{N}(\mu_r(x)\sigma_\theta(x) + \mu_\theta(x), \sigma_r(x)\sigma_\theta(x))$$

Using this predictive distribution, we can select the hyperparameter configuration maximizing the Expected Improvement (EI) (Mockus et al., 1978) of $g(x)$. The EI can then be defined in closed form as

$$\text{EI}(x) = \mathbf{E}[\max(0, g(x_{\min}) - \hat{g}(x))]$$
$$= \sigma^2(x)(v(x)\Phi(v(x)) + \phi(v(x))), \quad \text{where } v(x) := \frac{\mu_g(x) - g(x_{\min})}{\sigma_g^2(x)},$$

with $\Phi$ and $\phi$ being the CDF and PDF of the standard normal, respectively. When no observations are available, the empirical CDF $\tilde{F}$ is not defined. Therefore, we warm-start the optimization on the new task by sampling a set of $N_0 = 5$ hyperparameter configurations via Thompson sampling, as described above. Pseudo-code is given in Algorithm 1.

---

**Algorithm 1** Gaussian Copula process (CGP)

---

Learn the parameters $\theta$ of $\mu_\theta(x)$ and $\sigma_\theta(x)$ on hold-out evaluations $\mathcal{D}^M$ by minimizing equation 1.
Sample an initial set of evaluations $\mathcal{D} = \{(x_i, f(x_i))\}_{i=1}^{N_0}$ via Thompson sampling 2.
**while** Has budget **do**
    Fit the GP surrogate $\hat{r}$ to the observations $\{(x, \frac{\psi(y) - \mu_\theta(x)}{\sigma_\theta(x)}) \mid (x, y) \in \mathcal{D}\}$
    Sample $N$ candidate hyperparameters $x_1, \ldots, x_N$ from the search space
    Compute the hyperparameter $x_i$ where $i = \arg\max_i \text{EI}(x_i)$
    Evaluate $y_i = f(x_i)$ and update $\mathcal{D} = \mathcal{D} \cup \{(x_i, y_i)\}$.
**end while**

---

### 4.3 OPTIMIZING MULTIPLE OBJECTIVES

In addition to optimizing the accuracy of a black-box function, it is often desirable to optimize its runtime or memory consumption. For instance, given two hyperparameters with the same expected error, the one requiring fewer resources is preferable. For tasks where runtime is available, we use both runtime and error objectives by averaging in the transformed space, e.g., we set $z(x) = \frac{1}{2}(z^{\text{error}}(x) + z^{\text{time}}(x))$, where $z^{\text{error}}(x) = \psi(f^{\text{error}}(x))$ and $z^{\text{time}}(x) = \psi(f^{\text{time}}(x))$ denote the transformed error and time observations, respectively. This allows us to seamlessly optimize for time and error when running HPO, so that the cheaper hyperparameter is favored when two hyperparameters lead to a similar expected error. Notice many existing multi-objective methods can potentially be combined with our Copula transformation as an extension, which we believe is an interesting venue for future work.

| tasks | # datasets | # hyperparameters | # evaluations per dataset | available objectives |
|---|---|---|---|---|
| DeepAR | 11 | 6 | $\sim 220$ | quantile loss, time |
| FCNET | 4 | 9 | 62208 | MSE, time |
| XGBoost | 9 | 9 | 5000 | 1-AUC |

Table 1: A summary of the three HPO problems we consider.

## 5 EXPERIMENTS

We considered the problem of tuning three algorithms on multiple datasets: `XGBoost` (Chen & Guestrin, 2016), a 2-layer feed-forward neural network (`FCNET`) (Klein & Hutter, 2019), and the RNN-based time series prediction model proposed in Salinas et al. (2017) (`DeepAR`). We tuned `XGBoost` on 9 `libsvm` datasets (Chang & Lin, 2011) to minimize $1-$AUC, and `FCNET` on 4 datasets from Klein & Hutter (2019) to minimize the test mean squared error. As for `DeepAR`, the evaluations were collected on the data provided by GluonTS (Alexandrov et al., 2019), consisting of 6 datasets from the M4-competition (Makridakis et al., 2018) and 5 datasets used in Lai et al. (2017), and the goal is to minimize the quantile loss. Additionally, for `DeepAR` and `FCNET` the runtime to evaluate each hyperparameter configuration was available, and we ran additional experiments exploiting this objective. More details on the HPO setup are in Table 1, and the search spaces of the three problems is in Table 4 of the appendix. Lookup tables are used as advocated in Eggensperger et al. (2012), more details and statistics can be found in the appendix.

We compare against a number of baselines. We consider random search and GP-based BO as two of the most popular HPO methods. As a transfer learning baseline, we consider warm-start GP (Feurer et al., 2015), using the best-performing evaluations from all the tasks to warm start the GP on the target task (`WS GP best`). As an extension of `WS GP best`, we apply standardization on the objectives of the evaluations for every task and then use all of them to warm start the GP on the target task (`WS GP all`). We also compare against two recently published transfer learning methods for HPO: `ABLR` (Perrone et al., 2018) and a search space-based transfer learning method (Perrone et al., 2019). `ABLR` is a transfer learning approach consisting of a shared neural network across tasks on top of which lies a Bayesian linear regression layer per task. Finally, Perrone et al. (2019) transfers information by fitting a bounding box to contain the best hyperparameters from each previous task, and applies random search (`Box RS`) or GP-based BO (`Box GP`) in the learned search space.

We assess the transfer learning capabilities of these methods in a leave-one-task-out setting: we sequentially leave out one dataset and then aggregate the results for each algorithm. The performance of each method is first averaged over 30 replicates for one dataset in a task, and the relative improvements over random search are computed on every iteration for that dataset. The relative improvement for an optimizer (`opt`) is defined by $(y_{random} - y_{opt})/y_{random}$, which is upper bounded by $100\%$. Notice that all the objectives $y$ are in $\mathbb{R}^+$. By computing the relative improvements, we can aggregate results across all datasets for each algorithm. Finally, for all copula-based methods, we learn the mapping to copulas via a 3-layer MLP with 50 units per layer, optimized by ADAM with early-stopping.

### 5.1 ABLATION STUDY

To give more insight into the components of our method, we perform a detailed ablation study to investigate the choice of the MLP and compare the copula estimation to simple standardization.

**Choice of copula estimators** For copula-based methods, we use an MLP to estimate the output. We first compare to other possible options, including a linear model and a $k$-nearest neighbor estimator in a leave-one-out setting: we sequentially take the hyperparameter evaluations of one dataset as test set and use all evaluations from the other datasets as a training set. We report the RMSE in Table 5 of the appendix when predicting the error of the blackbox. From this table, it is clear that MLP tends to be the best performing estimator among the three. In addition, a low RMSE indicates that the task is close to the prior that we learned on all the other tasks, and we should thus expect transfer learning methods to perform well. As shown later by the BO experiments, `FCNET` has the lowest RMSE among the three algorithms, and all transfer learning methods indeed perform much better than single-task approaches.

**Homoskedastic and Heteroskedastic noise** The proposed Copula estimator (MLP) uses heteroskedastic noise for the prior. We now compare it to a homoskedastic version where we only estimate the mean. The results are summarized in Table 2 where average relative improvements over random search across all the iterations and replicates are shown. It is clear that heteroskedasticity tends to help on most datasets.

**Copula transformation and standardization** In our method, we map objectives to be normally distributed in two steps: first we apply the probability integral transform, followed by a Copula transform using the inverse CDF of a Gaussian. To demonstrate the usefulness of such transformation, we compare it to a simple standardization of the objectives where mean and std are computed on each datasets separately. Results are reported in Table 2. It is clear that standardization performs significantly worse than the Copula transformation, indicating that it is not able to address the problem of varying scale and noise levels across tasks. Note that the relative improvement objective is not lower bounded, so that when random search finds very small values the scale of relative improvement can be arbitrary large (such as for the `Protein` dataset in `FCNET`).

| task | dataset | TS std | CTS Ho | CTS He | GP std | CGP Ho | CGP He |
|------|---------|--------|--------|--------|--------|--------|--------|
| DeepAR | electricity | -13.2 | 0.3 | **0.8** | -15.3 | 0.4 | **0.8** |
| | exchange-rate | -127.2 | 1.8 | 2.9 | -130.6 | 3.0 | **3.3** |
| | m4-Daily | -58.0 | 1.0 | 1.1 | -107.3 | **1.4** | **1.4** |
| | m4-Hourly | -98.6 | -0.8 | -0.8 | -94.7 | 0.7 | **3.0** |
| | m4-Monthly | -24.2 | 0.3 | 0.6 | -19.2 | 0.9 | **1.0** |
| | m4-Quarterly | -15.6 | 0.5 | 0.8 | -11.8 | 0.8 | **1.0** |
| | m4-Weekly | -96.1 | 0.2 | 0.4 | -81.9 | 0.3 | **0.6** |
| | m4-Yearly | -14.0 | 0.4 | 0.7 | -13.6 | 0.8 | **1.1** |
| | solar | -14.1 | 0.4 | 0.5 | -8.7 | 0.8 | **1.1** |
| | traffic | -17.3 | 0.3 | 0.0 | -7.4 | **0.7** | 0.5 |
| | wiki-rolling | -4.5 | 0.3 | 0.3 | -4.6 | 0.4 | **0.5** |
| FCNet | naval | -20602.7 | 72.0 | 78.9 | -4368.4 | 81.7 | **82.3** |
| | parkinsons | -78.6 | 27.8 | 29.6 | -96.1 | **42.1** | 38.8 |
| | protein | -18.8 | 5.4 | 6.6 | -9.6 | 8.1 | **8.3** |
| | slice | -870.6 | 46.3 | 53.5 | 14.2 | 58.5 | **58.9** |
| XGBoost | a6a | -0.7 | 0.0 | -0.1 | -0.3 | **0.2** | **0.2** |
| | australian | -50.8 | 0.2 | 0.4 | -53.4 | 3.4 | **3.7** |
| | german.numer | -12.2 | 0.6 | 0.5 | -12.7 | 0.6 | **0.7** |
| | heart | -70.1 | -0.1 | 0.9 | -129.0 | 3.9 | **5.1** |
| | ijcnn1 | -38.8 | 1.7 | 3.0 | -11.7 | **5.9** | 5.8 |
| | madelon | -37.9 | -0.4 | -0.4 | -16.2 | **4.9** | 4.2 |
| | spambase | -30.2 | 0.6 | -0.9 | -16.9 | **2.6** | 1.0 |
| | svmguide1 | -28.3 | 0.8 | -0.3 | -17.0 | 1.2 | 1.2 |
| | w6a | 0.6 | 0.9 | 0.5 | 2.7 | **4.0** | 3.4 |

Table 2: Relative improvements over random search. `TS std` and `GP std` respectively using a simple standardization instead of the copula transformation. Ho and He stand for Homoskedastic and Heteroskedastic noise.

## 5.2 Results

We now compare the proposed methods to other HPO baselines. The results on using only the error information are shown first followed by the results using both time and error information.

**Results using only error information** We start by studying the setting where only error objectives are used to learn the copula transformation. Within each task, we first aggregate 30 replicates for each method to compute the relative improvement over random search at every iteration, and then average the results across all iterations. The results are reported in Table 3, showing that `CGP` is the best method for almost every task except `XGBoost`. In `XGBoost`, there are several tasks on which methods without transfer learning perform quite well. This is not surprising as we observe in an ablation study on copula estimators (see Table 5 in the appendix) that some tasks in `XGBoost` have relatively high test errors, implying that the transferred prior will not help. In those tasks, `CGP` is usually the runner-up method after standard `GP`. We also report the results at iteration 10, 50 and 100 in the Tables 7, 8 and 9 in the appendix where we observe `CGP` and `Box RS` are the most competitive methods at 10th iteration but at 100 iteration, `CGP` is clearly the best transfer learning method. This highlights the advantage of being adaptive to the target task of our method while making effective transfer in the beginning.

| task | dataset | ABLR | Box GP | Box RS | CGP | CTS | GP | WS GP all | WS GP best |
|---|---|---|---|---|---|---|---|---|---|
| DeepAR | electricity | -2.7 | -2.1 | **0.8** | **0.8** | **0.8** | 0.2 | -2.4 | -2.3 |
| | exchange-rate | 0.5 | -0.2 | 2.1 | **3.3** | 2.9 | 0.6 | 0.6 | 0.3 |
| | m4-Daily | -1.2 | -1.0 | 0.1 | **1.4** | 1.1 | -0.1 | -0.9 | -0.7 |
| | m4-Hourly | -18.0 | -12.1 | -2.8 | **3.0** | -0.8 | -1.3 | -9.8 | -11.7 |
| | m4-Monthly | -0.7 | 0.2 | 0.6 | **1.0** | 0.6 | 0.2 | 0.2 | 0.6 |
| | m4-Quarterly | -0.4 | -0.1 | 0.6 | **1.0** | 0.8 | 0.1 | 0.1 | 0.0 |
| | m4-Weekly | -3.8 | -3.2 | 0.5 | **0.6** | 0.4 | 0.1 | -2.9 | -3.2 |
| | m4-Yearly | -0.1 | 0.2 | 0.3 | **1.1** | 0.7 | 0.5 | -0.2 | 0.3 |
| | solar | -0.3 | 0.3 | 0.8 | **1.1** | 0.5 | 0.3 | -0.5 | -0.1 |
| | traffic | -1.3 | -0.7 | **0.5** | **0.5** | 0.0 | 0.2 | -0.4 | -0.3 |
| | wiki-rolling | 0.1 | 0.1 | 0.2 | **0.5** | 0.3 | 0.1 | 0.3 | 0.0 |
| FCNet | naval | 57.1 | 70.6 | 80.6 | **82.3** | 78.9 | -64.3 | 61.2 | 65.9 |
| | parkinsons | 14.1 | 27.1 | 27.3 | **38.5** | 29.4 | 20.2 | 19.9 | 27.6 |
| | protein | 0.3 | 6.8 | 5.8 | **8.3** | 6.6 | 3.3 | 6.3 | 5.9 |
| | slice | 1.8 | 37.1 | 48.3 | **58.7** | 53.3 | 21.4 | 41.9 | 35.9 |
| XGBoost | a6a | -0.1 | 0.1 | 0.1 | **0.2** | -0.1 | **0.2** | -0.1 | -0.1 |
| | australian | -1.2 | 1.4 | 3.1 | **3.7** | 0.4 | 1.7 | 1.5 | 0.8 |
| | german.numer | -1.3 | 0.2 | **1.3** | 0.7 | 0.5 | -0.3 | **1.3** | 0.2 |
| | heart | 1.5 | 1.5 | 2.3 | **5.1** | 0.9 | 2.8 | -2.1 | 4.4 |
| | ijcnn1 | -8.1 | 3.1 | 5.0 | **5.8** | 3.0 | 4.1 | 3.4 | 4.4 |
| | madelon | 2.4 | 3.5 | 1.7 | 4.2 | -0.4 | **4.4** | 0.3 | 1.4 |
| | spambase | -2.9 | 0.9 | 0.0 | 1.0 | -0.9 | **1.8** | -0.3 | -0.5 |
| | svmguide1 | -3.2 | 0.8 | **1.7** | 1.2 | -0.3 | 1.1 | 0.4 | 0.4 |
| | w6a | 0.8 | 1.1 | -2.5 | 3.4 | 0.5 | **3.8** | -0.8 | 0.9 |

Table 3: Relative improvements over random search averaged over all the iterations. The best methods are highlighted in bold.

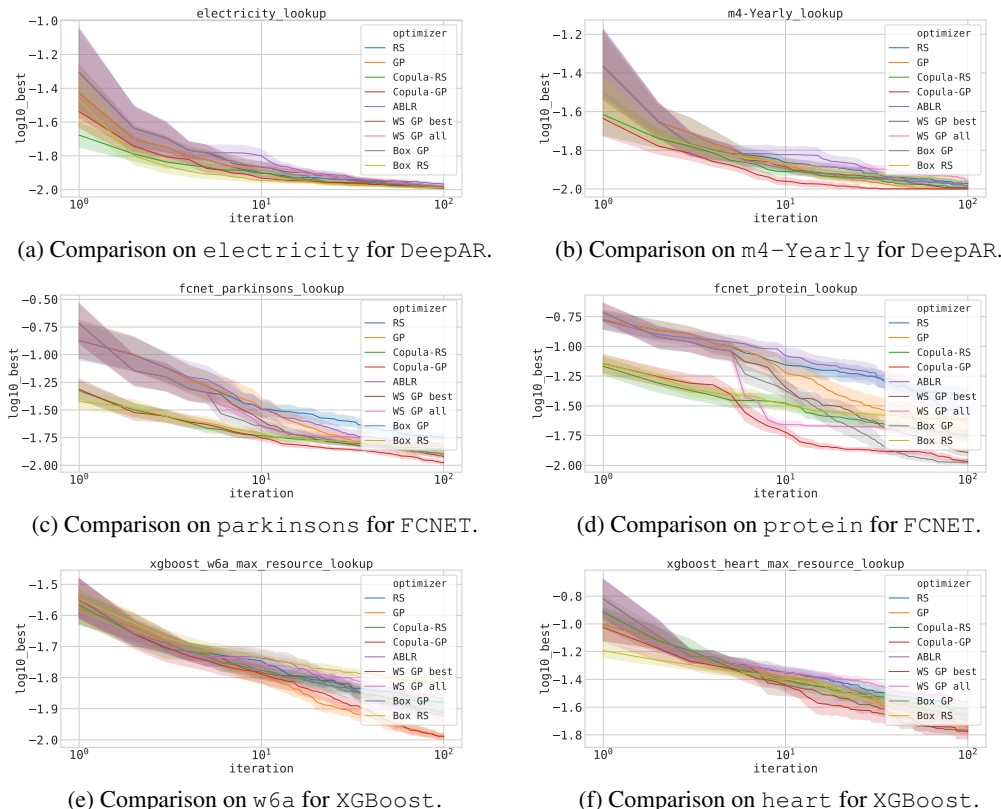

(a) Comparison on `electricity` for `DeepAR`.

(b) Comparison on `m4-Yearly` for `DeepAR`.

(c) Comparison on `parkinsons` for `FCNET`.

(d) Comparison on `protein` for `FCNET`.

(e) Comparison on `w6a` for `XGBoost`.

(f) Comparison on `heart` for `XGBoost`.

Figure 1: Results using only error information, with the current optimum plotted against the number of used evaluations.

We also show results on two example datasets from each algorithm in Figure 1, reporting confidence intervals obtained via bootstrap. Note that the performance of the methods in the examples for `DeepAR` and `XGBoost` exhibit quite high variations, especially in the beginning of the BO. We

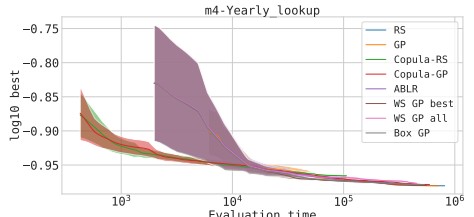 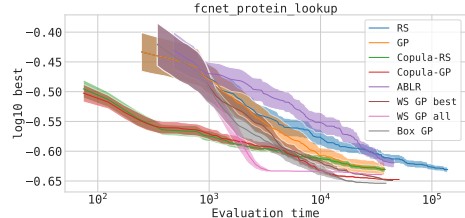

(a) Comparison on dataset `m4-Yearly` over time.     (b) Comparison on dataset `protein` over time.

Figure 2: Two example tasks using both error and time information, with the current optimum plotted against time.

conjecture this is due to an insufficient number of evaluations in the lookup tables. Nevertheless, the general trend is that `CTS` and `CGP` outperform all baselines, especially in the beginning of the BO. It can also be observed that `CGP` performs at least on par with the best method at the end of the BO. `Box RS` is also competitive at the beginning, but as expected fails to keep its advantage toward the end of the BO.

**Results using both error and time information**    We then studied the ability of the copula-based approaches to transfer information from multiple objectives. Notice it is possible to combine Copula transformation with other multi-objective BO methods and we will leave this direction as future work. We show two example tasks in `DeepAR` and `FCNET` in Figure 2, where we fix the total number of iterations and plot performance against time with 2 standard error. To obtain distributions over seeds for one method, we only consider the time range where 20 seeds are available ,which explains why methods can start and end at different times. With the ability to leverage training time information, the copula-based approaches have a clear advantage over all baselines, especially at the beginning of the optimization.

We also report aggregate performance over all the tasks in Table 6 in the appendix. Due to the different methods finishing the optimization at different times, we only compare them up to the time taken by the fastest method. For each method we first compute an average over 30 replicates, then compute the relative improvement over random search, and finally average the results across all time points. The copula based methods converge to a good hyperparameter configuration significantly faster than all the considered baselines. Note that we obtain similar results as for Hyperband-style methods (Li et al., 2016), where the optimization can start much earlier than standard HPO, with the key difference that we only require a single machine.

## 6 CONCLUSIONS

We introduced a new class of methods to accelerate hyperparameter optimization by exploiting evaluations from previous tasks. The key idea was to leverage a semi-parametric Gaussian Copula prior, using it to account for the different scale and noise levels across tasks. Experiments showed that we considerably outperform standard approaches to BO, and deal with heterogeneous tasks more robustly compared to a number of transfer learning approaches recently proposed in the literature. Finally, we showed that our approach can seamlessly combine multiple objectives, such as accuracy and runtime, further speeding up the search of good hyperparameter configurations.

A number of directions for future work are open. First, we could combine our Copula-based HPO strategies with Hyperband-style optimizers (Li et al., 2016). In addition, we could generalize our approach to deal with settings in which related problems are not limited to the same algorithm run over different datasets. This would allow for different hyperparameter dimensions across tasks, or perform transfer learning across different black-boxes.

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

# A    APPENDIX

---

**Algorithm 2** Copula Thompson sampling (CTS)

---
Learn the parameters $\theta$ of $\mu_\theta(x)$ and $\sigma_\theta(x)$ on hold-out evaluations $\mathcal{D}^M$ by minimizing equation 1.
**while** Has budget **do**
    Sample $N$ candidate hyperparameters $x_1, \ldots, x_N$ from the search space
    Draw $\tilde{z}_i \sim \mathcal{N}(\mu_\theta(x_i), \sigma_\theta(x_i))$ for $i = 1, \ldots, N$
    Evaluate $f(x_i)$ where $i = \arg\min_i \tilde{z}_i$
**end while**

---

## A.1    LOOKUP TABLES

To speed up experiments we used a lookup table approach advocated in Eggensperger et al. (2012) which proposed to use an extrapolation model built on pre-generated evaluations to limit the number of blackbox evaluations, thus saving a significant amount of computational time. However, the extrapolation model can introduce noise and lead to inconsistencies compared to using real blackbox evaluations. As a result, in this work we reduced BO to the problem of selecting the next hyperparameter configurations from a fixed set that has been evaluated in advance, so that no extrapolation error is introduced.

All evaluations were obtained by querying each algorithm at hyperparameters sampled (log) uniformly at random from their search space as described in Table 4. The CDF on the error objectives is given in Figure 3.

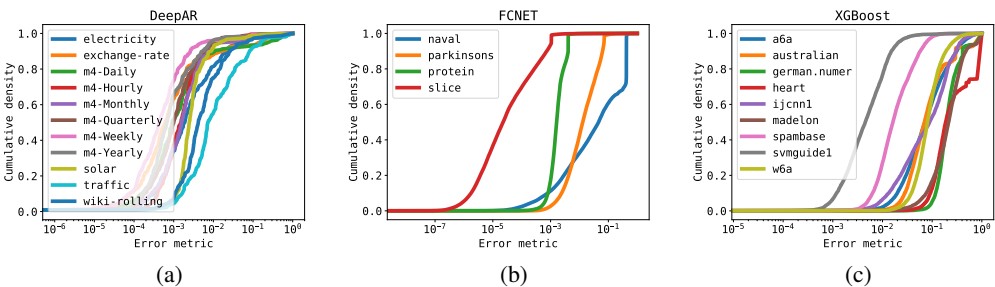

Figure 3: CDF of error metrics for the three tasks. Every line represent one dataset. The metrics are scaled first by min-max scaling and the x-axis is log scaled.

**Results on different iterations.**    We plot the improvement over random research for all the methods at iteration 10, 50 and 100 at Table 7, 8 and 9, respectively. In short, at 10th iteration, transfer learning methods, especially our CGP and Box RS, performed much better than GP. But, when looking at results at 50 and 100 iterations, CGP outperforms clearly all other transfer methods because of its improved adaptivity.

**More details on prior MLP architecture.**    The MLP used to regress $\mu_\theta$ and $\sigma_\theta$ consists of 3 layers with 50 nodes, each with a dropout layer set to 0.5. The learning rate is set to 0.01, batch size to 64 and we optimize over 100 gradient updates 3 times, lowering the learning rate by 10 each time.

| tasks | hyperparameter | search space | type | scale |
|---|---|---|---|---|
| DeepAR | # layers | $[1, 5]$ | integer | linear |
| | # cells | $[10, 120]$ | integer | linear |
| | learning rate | $[10^{-4}, 0.1]$ | continuous | log10 |
| | dropout rate | $[10^{-2}, 0.5]$ | continuous | log10 |
| | context_length_ratio | $[10^{-1}, 4]$ | continuous | log10 |
| | # bathes per epoch | $[10, 10^4]$ | integer | log10 |
| XGBoost | num_round | $[2, 2^9]$ | integer | log2 |
| | eta | $[0, 1]$ | continuous | linear |
| | gamma | $[2^{-20}, 2^6]$ | continuous | log2 |
| | min_child_weight | $[2^{-8}, 2^6]$ | continuous | log2 |
| | max_depth | $[2, 2^7]$ | integer | log2 |
| | subsample | $[0.5, 1]$ | continuous | linear |
| | colsample_bytree | $[0.3, 1]$ | continuous | linear |
| | lambda | $[2^{-10}, 2^8]$ | continuous | log2 |
| | alpha | $[2^{-20}, 2^8]$ | continuous | log2 |
| FCNET | initial_lr | $\{0.005, 0.001, 0.05, 0.01, 0.05, 0.1\}$ | categorical | - |
| | batch_size | $\{8, 16, 32, 64\}$ | categorical | - |
| | lr_schedule | $\{\text{cosine, fix}\}$ | categorical | - |
| | activation layer 1 | $\{\text{relu, tanh}\}$ | categorical | - |
| | activation layer 2 | $\{\text{relu, tanh}\}$ | categorical | - |
| | size layer 1 | $\{16, 32, 64, 128, 256, 512\}$ | categorical | - |
| | size layer 2 | $\{16, 32, 64, 128, 256, 512\}$ | categorical | - |
| | dropout layer 1 | $\{0.0, 0.3, 0.6\}$ | categorical | - |
| | dropout layer 2 | $\{0.0, 0.3, 0.6\}$ | categorical | - |

Table 4: A summary of the search spaces for the three algorithms.

| task | dataset | KNN_10 | KNN_20 | KNN_5 | Linear | MLP |
|---|---|---|---|---|---|---|
| DeepAR | electricity | 0.831 | 0.831 | 0.831 | 0.801 | 0.740 |
| | exchange-rate | 0.842 | 0.842 | 0.842 | 0.783 | 0.780 |
| | m4-Daily | 0.804 | 0.804 | 0.804 | 0.792 | 0.776 |
| | m4-Hourly | 0.960 | 0.960 | 0.960 | 0.948 | 0.884 |
| | m4-Monthly | 0.783 | 0.783 | 0.783 | 0.762 | 0.750 |
| | m4-Quarterly | 0.868 | 0.868 | 0.868 | 0.792 | 0.773 |
| | m4-Weekly | 0.776 | 0.776 | 0.776 | 0.754 | 0.733 |
| | m4-Yearly | 0.844 | 0.844 | 0.844 | 0.785 | 0.759 |
| | solar | 0.963 | 0.963 | 0.963 | 0.875 | 0.812 |
| | traffic | 0.885 | 0.885 | 0.885 | 0.850 | 0.829 |
| | wiki-rolling | 0.904 | 0.904 | 0.904 | 0.868 | 0.826 |
| FCNet | naval | 0.509 | 0.509 | 0.509 | 0.602 | 0.491 |
| | parkinsons | 0.571 | 0.571 | 0.571 | 0.736 | 0.571 |
| | protein | 0.505 | 0.505 | 0.505 | 0.607 | 0.497 |
| | slice | 0.564 | 0.564 | 0.564 | 0.559 | 0.555 |
| XGBoost | a6a | 1.091 | 1.091 | 1.091 | 1.067 | 1.040 |
| | australian | 0.827 | 0.827 | 0.827 | 0.873 | 0.758 |
| | german.numer | 0.900 | 0.900 | 0.900 | 0.891 | 0.820 |
| | heart | 0.818 | 0.818 | 0.818 | 0.793 | 0.702 |
| | ijcnn1 | 0.951 | 0.951 | 0.951 | 0.936 | 0.917 |
| | madelon | 0.908 | 0.908 | 0.908 | 0.887 | 0.834 |
| | spambase | 0.931 | 0.931 | 0.931 | 0.950 | 0.818 |
| | svmguide1 | 0.849 | 0.849 | 0.849 | 0.912 | 0.798 |
| | w6a | 1.039 | 1.039 | 1.039 | 1.054 | 1.003 |

Table 5: RMSE comparison for prior estimators when predicting the blackbox error given its parameters.

| task | dataset | ABLR | Box GP | CGP | CTS | GP | WS GP all | WS GP best |
|------|---------|------|--------|-----|-----|----|-----------|-----------|
| DeepAR | electricity | -19.7 | -17.2 | 10.0 | 10.3 | -0.3 | -18.5 | -17.0 |
| | exchange-rate | 7.4 | 6.4 | 38.7 | 39.1 | 0.2 | -0.3 | 7.7 |
| | m4-Daily | -17.5 | -15.7 | 20.1 | 20.6 | -1.7 | -16.9 | -14.8 |
| | m4-Hourly | -103.6 | -99.7 | 30.9 | 29.8 | 3.0 | -102.0 | -102.0 |
| | m4-Monthly | 3.7 | 6.7 | 5.0 | 4.6 | -0.6 | 6.5 | 7.6 |
| | m4-Quarterly | -10.3 | -9.6 | 0.7 | -0.4 | -0.6 | -9.2 | -9.6 |
| | m4-Weekly | -26.7 | -24.6 | 0.2 | -1.4 | -0.6 | -23.0 | -23.5 |
| | m4-Yearly | -3.8 | -3.1 | 2.2 | 2.1 | -0.1 | -3.7 | -2.9 |
| | solar | 1.6 | 4.0 | 5.6 | 5.6 | 0.1 | 3.4 | 3.3 |
| | traffic | -36.7 | -34.5 | 3.6 | 3.8 | 0.4 | -33.7 | -33.6 |
| | wiki-rolling | -1.1 | -0.9 | 2.3 | 2.5 | 0.1 | -0.3 | -1.0 |
| FCNet | naval | -80.6 | 69.4 | 90.1 | 89.5 | -150.0 | 45.7 | 58.8 |
| | parkinsons | -38.5 | 30.3 | 53.2 | 46.4 | 22.3 | 27.0 | 26.9 |
| | protein | -6.9 | 10.1 | 12.7 | 10.8 | 4.1 | 12.6 | 7.4 |
| | slice | -14.5 | 6.1 | 83.3 | 80.9 | 22.4 | 8.1 | 8.4 |

Table 6: Relative improvements over random search averaged over time.

| task | dataset | ABLR | Box GP | Box RS | Copula-GP | Copula-TS | GP | WS GP all | WS GP best |
|------|---------|------|--------|--------|-----------|-----------|----|-----------|-----------|
| DeepAR | electricity | -4.1 | -1.2 | **1.4** | *1.0* | 0.1 | -0.1 | -1.3 | -1.4 |
| | exchange-rate | -2.0 | 4.5 | *5.7* | 3.6 | **5.8** | 0.9 | 4.8 | 3.4 |
| | m4-Daily | -3.4 | -1.3 | **1.0** | 0.5 | 0.6 | 0.3 | -1.1 | 0.6 |
| | m4-Hourly | -1.9 | 2.9 | 4.7 | 4.7 | *5.0* | **5.4** | 3.1 | 3.9 |
| | m4-Monthly | -1.5 | 1.0 | *1.6* | *1.6* | 1.5 | 0.2 | 1.5 | **1.7** |
| | m4-Quarterly | -1.3 | 0.0 | 0.0 | **1.0** | *0.7* | 0.0 | *0.7* | -0.1 |
| | m4-Weekly | -2.6 | -0.1 | **2.1** | 1.1 | *1.8* | 0.2 | 0.8 | 1.0 |
| | m4-Yearly | -0.8 | 0.5 | 0.1 | **1.3** | *0.7* | 0.3 | 0.1 | 0.3 |
| | solar | -1.3 | 0.5 | **1.8** | *1.4* | *1.4* | 0.6 | 0.3 | 0.0 |
| | traffic | -3.0 | -1.1 | **1.0** | *0.2* | -0.1 | 0.1 | -0.5 | -0.5 |
| | wiki-rolling | 0.2 | 0.5 | 0.6 | *0.8* | 0.6 | 0.1 | **1.0** | 0.4 |
| FCNet | naval | 79.8 | 87.9 | *95.7* | **95.8** | 94.9 | -841.0 | 63.2 | 71.0 |
| | parkinsons | 5.9 | 32.7 | 42.9 | **46.9** | *45.1* | -3.3 | 34.9 | 24.6 |
| | protein | -2.6 | 5.9 | *8.7* | **11.4** | 8.3 | 0.8 | 10.9 | 5.7 |
| | slice | -16.7 | 59.5 | 74.0 | **80.0** | *78.9* | -42.5 | 59.7 | 59.7 |
| XGBoost | a6a | -0.5 | -0.1 | **0.0** | *-0.1* | -0.2 | -0.2 | -0.2 | -0.3 |
| | australian | -7.9 | 1.6 | **5.3** | *3.1* | 1.2 | 1.2 | -1.5 | 0.5 |
| | german.numer | -0.7 | 0.6 | **1.6** | *0.9* | 0.6 | -0.2 | 0.7 | -0.3 |
| | heart | -0.8 | 2.1 | 2.5 | **4.2** | 2.8 | 1.5 | -0.3 | *3.8* |
| | ijcnn1 | -13.9 | -0.4 | **9.1** | *8.0* | 5.2 | 2.9 | 5.9 | 6.6 |
| | madelon | -1.4 | 2.1 | **4.3** | *3.0* | -1.2 | 1.0 | 1.2 | 4.3 |
| | spambase | -5.5 | -2.1 | **0.2** | -2.7 | -1.6 | -1.9 | -0.6 | *-0.5* |
| | svmguide1 | -2.9 | 0.9 | **2.0** | *1.2* | 0.2 | 0.0 | 0.1 | *1.2* |
| | w6a | -0.9 | 1.6 | -0.7 | *2.1* | 1.7 | **2.2** | 0.5 | *2.1* |

Table 7: Relative improvements over random search at iteration 10.

| task | dataset | ABLR | Box GP | Box RS | Copula-GP | Copula-TS | GP | WS GP all | WS GP best |
|------|---------|------|--------|--------|-----------|-----------|----|-----------|-----------|
| DeepAR | electricity | -0.6 | -0.2 | **0.3** | *0.1* | *0.1* | *0.1* | -0.6 | -0.4 |
| | exchange-rate | *1.8* | -0.2 | -0.1 | **2.1** | 1.3 | 0.6 | 1.2 | 1.0 |
| | m4-Daily | -0.2 | -0.2 | *0.2* | **0.6** | -0.2 | 0.1 | 0.0 | -0.2 |
| | m4-Hourly | -10.3 | -4.2 | -4.7 | **1.9** | -3.8 | -4.1 | *-2.0* | -3.9 |
| | m4-Monthly | -0.3 | 0.2 | -0.1 | **0.8** | 0.2 | 0.4 | 0.1 | *0.5* |
| | m4-Quarterly | -0.5 | -0.1 | *0.2* | **0.4** | 0.1 | *0.2* | 0.1 | 0.0 |
| | m4-Weekly | -0.1 | 0.3 | 0.3 | **1.0** | 0.2 | 0.4 | *0.5* | 0.1 |
| | m4-Yearly | 0.1 | 0.2 | -0.1 | **0.7** | 0.4 | *0.5* | -0.3 | 0.3 |
| | solar | 0.2 | 0.4 | 0.4 | **1.0** | 0.4 | *0.5* | -0.7 | -0.2 |
| | traffic | -0.5 | -0.4 | 0.0 | **0.3** | -0.2 | **0.3** | 0.1 | 0.2 |
| | wiki-rolling | -0.2 | -0.1 | -0.2 | **0.1** | -0.1 | *0.0* | *0.0* | -0.5 |
| FCNet | naval | 59.6 | 72.7 | *80.4* | **81.5** | 77.9 | 46.2 | 64.4 | 70.9 |
| | parkinsons | 16.9 | 26.1 | 23.1 | **33.5** | 24.4 | 24.9 | 19.5 | *27.5* |
| | protein | 0.0 | **7.0** | 4.2 | *6.8* | 5.1 | 3.2 | 5.3 | 5.8 |
| | slice | 4.5 | 39.2 | 43.3 | **55.3** | *48.0* | 34.2 | 44.9 | 38.8 |
| XGBoost | a6a | -0.1 | 0.1 | 0.1 | **0.2** | -0.1 | **0.2** | -0.1 | -0.1 |
| | australian | -0.9 | 1.3 | 2.0 | **4.3** | -0.3 | 1.8 | 2.1 | 0.0 |
| | german.numer | -1.5 | 0.1 | *1.0* | 0.4 | 0.5 | -0.5 | **1.5** | 0.3 |
| | heart | 2.6 | 2.5 | 2.3 | *5.1* | 1.8 | 3.7 | -0.8 | **6.6** |
| | ijcnn1 | -8.1 | 4.2 | 4.5 | **7.5** | 3.2 | *5.8* | 4.2 | 5.6 |
| | madelon | 3.6 | 3.5 | 0.9 | *4.3* | -0.1 | **4.9** | 0.5 | 1.0 |
| | spambase | -2.8 | 1.5 | -0.5 | *1.6* | -0.6 | **2.2** | -0.1 | -0.4 |
| | svmguide1 | -3.1 | 0.9 | *1.2* | 1.1 | -0.3 | **1.3** | 0.6 | 0.7 |
| | w6a | 1.8 | 1.9 | -2.3 | *4.3* | 1.0 | **4.8** | -0.6 | 1.0 |

Table 8: Relative improvements over random search at iteration 50.

| task | dataset | ABLR | Box GP | Box RS | Copula-GP | Copula-TS | GP | WS GP all | WS GP best |
|------|---------|------|--------|--------|-----------|-----------|-----|-----------|------------|
| DeepAR | electricity | -0.5 | -0.2 | 0.0 | 0.0 | 0.0 | -0.1 | -0.2 | -0.2 |
| | exchange-rate | **1.0** | -0.4 | -0.4 | *0.8* | 0.1 | 0.3 | 0.4 | 0.6 |
| | m4-Daily | -0.2 | -0.5 | *-0.1* | **0.1** | *-0.1* | -0.3 | -0.2 | -0.3 |
| | m4-Hourly | -6.2 | -0.6 | -5.6 | **4.4** | -1.5 | 2.2 | *3.8* | -1.5 |
| | m4-Monthly | -0.1 | 0.1 | -0.2 | **0.3** | -0.1 | *0.2* | 0.1 | *0.2* |
| | m4-Quarterly | *0.0* | -0.1 | -0.1 | **0.1** | -0.1 | *0.0* | *0.0* | -0.1 |
| | m4-Weekly | 0.2 | 0.2 | 0.1 | **0.4** | 0.1 | 0.2 | 0.2 | 0.1 |
| | m4-Yearly | 0.2 | 0.1 | -0.1 | 0.3 | 0.3 | 0.3 | -0.1 | 0.3 |
| | solar | *0.3* | 0.0 | 0.2 | **0.7** | 0.1 | *0.3* | -0.4 | -0.3 |
| | traffic | -0.2 | -0.1 | 0.0 | **0.2** | -0.1 | **0.2** | 0.0 | 0.1 |
| | wiki-rolling | **0.0** | -0.2 | -0.3 | **0.0** | -0.1 | -0.1 | -0.1 | -0.2 |
| FCNet | naval | 41.1 | 58.2 | *63.0* | **66.0** | 60.8 | 26.1 | 54.1 | 51.5 |
| | parkinsons | 13.7 | 24.2 | 14.1 | **34.2** | 18.0 | 24.4 | 10.4 | *27.3* |
| | protein | 2.4 | **6.3** | 3.4 | **6.3** | 4.8 | 3.5 | 4.7 | 5.8 |
| | slice | -1.3 | 27.3 | 30.9 | **42.4** | *36.5* | 18.2 | 31.6 | 24.2 |
| XGBoost | a6a | 0.1 | 0.2 | 0.1 | **0.3** | 0.0 | **0.3** | -0.1 | -0.1 |
| | australian | 0.2 | 1.8 | 1.8 | **4.3** | 0.0 | *3.5* | 1.1 | 1.2 |
| | german.numer | -1.2 | 0.3 | **1.3** | 0.8 | 0.3 | -0.2 | *1.2* | 0.6 |
| | heart | 1.5 | 0.9 | -0.3 | *3.3* | -1.3 | 2.4 | -3.5 | **3.6** |
| | ijcnn1 | -2.7 | 3.1 | 2.4 | **4.8** | 1.6 | *4.0* | 1.7 | 2.8 |
| | madelon | 3.3 | *4.0* | -0.3 | *4.0* | -0.1 | **4.8** | 0.3 | 1.0 |
| | spambase | -1.6 | 1.4 | -0.7 | 2.2 | -1.4 | **3.0** | -0.7 | -0.9 |
| | svmguide1 | -2.7 | *1.2* | 1.1 | 0.8 | -0.3 | **1.7** | 0.8 | 0.2 |
| | w6a | 1.5 | 1.2 | -3.3 | **4.6** | 0.1 | **4.6** | -0.6 | 2.0 |

Table 9: Relative improvements over random search at iteration 100.

