# OpenReview forum: "A Copula approach for hyperparameter transfer learning"
_ICLR.cc/2020/Conference — Reject_

### Official Review · AnonReviewer1 · 2019-10-20
**Official Blind Review #1**

**Rating:** 1

**Review:**

The authors propose a new way of normalizing the labels of the meta-data. They propose a Thompson sampling strategy as a new hyperparameter optimization warmstarting strategy and an optimization method that leverages transfer learning.


The transfer learning baselines are surprisingly weak and show no improvement over the random search baseline for two of your tasks. You should consider stronger baselines. One interesting work is "Collaborative hyperparameter tuning" by Bardenet et al. which overcomes the problem of different scales by considering the problem as a ranking problem. You discussed further works in your related work. Another interesting work based on GPs is "Scalable Gaussian Process based Transfer Surrogates for Hyperparameter Optimization".
The warm-start GP currently seems to be your strongest baseline. However, I have doubts that it is implemented correctly. You say that you estimate the most similar dataset and then evaluate its best 100 hyperparameter configuration. First of all, I don't understand why you decided to choose 100 (DeepAR only has 220 configurations in total!). Second, this is not how this method works. Instead, you estimate the k most similar tasks and evaluate its best hyperparameter configuration. In your case k is upper bounded by 10 (number of tasks). This is notably smaller than 100 and gives more time to the Bayesian optimization which will likely improve your results.


You observe worse results on XGBoost and good ones on FCNet. You refer to Table 6 and connect it to the RMSE. This might be true but I have a much simpler explanation: the search space of FCNet is orders of magnitudes larger and it contains many configurations that lead to very high MSE and only few with low. Therefore, a random search will on average provide poor results where a warmstarted search will obtain decent results for the first iterations. XGBoost is less sensitive to hyperparameters such that the overall variance in the losses is smaller. Maybe you can provide some insights into the complexity of the optimization tasks (plot the distributions )and add it to the appendix?


Table 6 contains more datasets than Table 2. Why did you drop some tasks?


The aggregated results in Table 2 are nice but actually we are interested in the outcome after the search after a given budget. Can you add such a table?


I have few suggestions to improve the readability of the paper:
That there is a choice of copula estimators is mentioned at the very end of the paper. Can you add it to the section where you describe them first? In section 5.1 you already argue by referring to Table 6. However, Table 6 is explained first in section 5.2 which makes it hard to follow your argumentation.
You use the term "metric" to refer to objectives. This is confusing, you might consider changing this. You propose to use scalarization to address the multi-objective optimization problem. Why would an average of both objectives be the optimal solution? Does the unit you use to measure the time matter? What if you use machine learning algorithms that scale super-linear in the number of data points? How is this novel and why can't your baselines employ the same idea? A discussion of related work on autoML for multiple objectives is missing.
The second paragraph of section 5 is confusing. You cite some work, discuss it and then conclude that your setup is entirely different. Would any information be lost if you say that you precomputed the values?

**Experience Assessment:**

I have published in this field for several years.

**Review Assessment: Checking Correctness Of Derivations And Theory:**

I assessed the sensibility of the derivations and theory.

**Review Assessment: Checking Correctness Of Experiments:**

I carefully checked the experiments.

**Review Assessment: Thoroughness In Paper Reading:**

I read the paper thoroughly.

---

> ### Author Response · Authors · 2019-11-12
> **Response to Reviewer 1**
>
> We thank the reviewer for the two relevant references on related methods to achieve transfer learning. However, the two approaches differ importantly in our opinion. The first approach [1] fits a GP on observations of all tasks that are mapped through a rank estimator to avoid scale issues. We agree that the motivation for the ranking estimator is very similar: enable transfer learning by avoiding scaling issues. However, this approach is not applicable in our case as the runtime scales cubically in the number of observations, which is always greater than 50K. For this reason, [2] also claims that this baseline cannot be applied to a larger dataset. Furthermore, this approach cannot adapt to discrepancies of the current task as all observations are seen through an estimator fit on other tasks. For instance, a rank predictor always predicting zero will feed only zeros as observations to the GP, whereas our approach allows the posterior to deviate from the prior given new observations. Finally, the rank estimate is a single point prediction and uncertainty is not estimated from other tasks.
>
> Those issues motivated the second paper that you referenced [2], which proposes linear mixture of experts where the experts are GPs fitted on each dataset. To avoid numerical scale issues, the acquisition function is adapted to measure the weighted improvement on all tasks rather than just the current one. However, this approach only supports Gaussian noise, requires a non-standard acquisition function as well as hand-designed meta-features to estimate the mixture weights. In addition, it still requires fitting a GP on each dataset where the number of observations can be greater than 50K. Our parametric approach does not suffer from these shortcomings. We discussed these papers in the revision.
>
> [1] Collaborative hyperparameter tuning
>
> [2] Scalable Gaussian Process based Transfer Surrogates for Hyperparameter Optimization
>
> When comparing against warm-start GP, we took 100 evaluations to get a representative number of samples from the closest task without making the cubical scaling prohibitive. Following on the reviewer’s feedback, we ran additional experiments by taking the best evaluation from each related task to initialize the GP. The results confirm that our approach still outperforms this baseline, pointing to the benefits of mapping the tasks to a comparable space. We added these results to the paper.
>
> As suggested, we also checked the CDFs of the error metrics of these tasks and included the plots in the appendix. From the CDFs, we observe in XGBoost that more hyperparameter configurations lead to higher objective value than in FCNet. In addition, while the slice dataset has the largest proportion of good hyperparameters, it is not the case that the benefits of transfer learning is less evident; in fact, the improvement over random search is large and only second to the one observed on naval. As a result, we still believe that low RMSE are correlated with performance as our method works significantly better when the RMSE scores are low. The advantage of our method increases as the RMSE tends to zero, and the prior becomes perfect in the limit case.
>
>
> We also highlight that a key benefit of our method is the adaptivity to new tasks. As a result, when plain GP performs best on some datasets for XGBoost, our method is always at least the second best and beats all other transfer methods (Table 2).
>
> * "Table 6 contains more datasets than Table 2. Why did you drop some tasks?"
>
> We kept only public datasets in the final evaluation that were inadvertently left in this table.
>
> * "The aggregated results in Table 2 are nice but actually we are interested in the outcome after the search after a given budget. Can you add such a table?"
>
>  We included such tables at iteration 10, 50 and 100 in the appendix. In short, at the 10th iteration transfer learning methods, especially our CGP and Box RS, performed much better than GP. When looking at results at 50 and 100 iterations, CGP significantly outperforms all other transfer methods because of its improved adaptivity.
>
> * "A discussion of related work on autoML for multiple objectives is missing."
>
> One key novelty of our method is to use Copula transformation for transfer learning. This makes metrics in different units (such as time) comparable. While proposing another multi-objective BO method is out of the scope of the paper, we aimed to demonstrate how the Copula transformation can be seamlessly used in a multi-objective setting. Indeed, many existing multi-objective methods can potentially also be combined with our Copula transformation as an extension, which we believe is an interesting venue for future work. We made this point clearer in the paper.
>
> We thank the reviewer and have improved the readability of the paper with the proposed suggestions.

---

### Official Review · AnonReviewer2 · 2019-10-23
**Official Blind Review #2**

**Rating:** 3

**Review:**

This paper tackles the problem of black-box hyperparameter optimization when multiple related optimization tasks are available simultaneously, performing transfer learning between tasks. Different tasks correspond to different datasets and/or metrics. Gaussian copulas are used to synchronize the different scales of the tasks.

I have several reservations with this paper. First and foremost, it seems to be lacking a fair and trivial baseline (I will describe it below) that justifies the apparently unnecessary complicated path followed in this paper. Second, there are a few small incorrect or improperly justified technical details throughout the paper.


1) Mistaken/unjustified technical details:

- In equation 1, the last term seems to be constant. For each task, the function psi is not parametric, so its gradient is also not parametric and the input is the inverse of z, i.e., y, which is also fixed. So why is it included in the cost function? This sort of probabilistic renormalization is important in e.g. warped GPs because the transformation is parametric. In this case, I don't see the point. It can be treated as a normalization of the input data, prior to its probabilistic modeling.

- Before equation 1, the text says "by minimizing the Gaussian negative log-likelihood on the available evaluations (x, z)" But then, equation 1 is not the NLL on z but on y.

- In section 4.2 the authors model the residuals of the previous model using a powerful Matern-5/2 GP. Why modeling the residuals this way and not the observations themselves? The split of modeling between a parametric and non-parametric part is not justified.

- One of the main points of the variable changes is to normalize the scales of the different tasks. However, equations 1 adds together the samples of the different tasks (which, as pointed out by the authors might have different sizes). Even if the scales of the outputs are uniform, the different dataset sizes will bias the solutions towards larger datasets. Why would that be a good thing? This is not mentioned and doesn't seem correct: there should not be a connection between a dataset size and the prior influence of the corresponding task. In fact, this will have the same effect as if the cost had different scales for different tasks, which is precisely the problem that the authors are trying to avoid.


2) Trivial baseline

Given that the authors are trying to aggregate information about the optimal hyperparameters from several tasks, they should not compare with single-task approaches, but with the simplest way to combine all the tasks. For instance:
    a) Normalize the outputs of every task. This can be accomplished in the usual way by dividing by the standard deviation, or even better, by computing the fixed transform z = psi(y), separately for each task.
    b) Collect the z of all tasks and feed them into an existing GP black-box Bayesian optimizer.

This is a very simple way to get "transfer learning" and it's unclear that the extra complexities of this paper (copulas, changes of variable with proper renormalization when the transformation is parameter free, etc) are buying much else.


Minor improvements:

- Page 2: "is the output of a multi-layer perceptron (MLP) with d hidden nodes" Is d really the number of hidden nodes of the MLP? Or the number of outputs? Given that d is also the size of w, it seems it's actually the latter.

- Explain why the EI approach is used for the second model (with the GP), but not for the first model.

Edit after rebuttal:
“The term is not constant over z” -> Sure, it’s not constant over z. But z is constant. So the term is constant.

“The NLL is minimized in z and there is indeed no y in equation 1.” -> Sure, there’s no y in the equation, that’s correct. But it is still the NLL of y, and not the NLL of z.

About the new baseline: Instead of simply renormalizing using mean and standard deviation, I suggested above using the same z=psi(y) that is used in the paper for the normalization. Is that where the advantage of the proposed method is coming from?

"Note that this is orthogonal to the scale issues we focus on: larger tasks will have larger gradient contributions but the scaling we propose still allows us to learn tied parameters across tasks as their scales are made similar. " Both issues affect the scaling of the task, so I don't see how they can be orthogonal. Their scales are not made similar precisely because of the different sample sizes.


**Experience Assessment:**

I have read many papers in this area.

**Review Assessment: Checking Correctness Of Derivations And Theory:**

I assessed the sensibility of the derivations and theory.

**Review Assessment: Checking Correctness Of Experiments:**

I assessed the sensibility of the experiments.

**Review Assessment: Thoroughness In Paper Reading:**

I read the paper at least twice and used my best judgement in assessing the paper.

---

> ### Author Response · Authors · 2019-11-12
> **Response to reviewer 2**
>
> * "In equation 1, the last term seems to be constant. [...] So why is it included?"
>
> The term $\psi’(\psi^{-1}(z))$ is not constant over $z$ and is needed when performing the change of variable. It weights more the error (and gradients) occurring when the quantile function of $y$ is changing rapidly. In other words, error in regions where the quantile function changes rapidly should cost more compared to flat regions.
>
> * "Before equation 1, the text says 'by minimizing the Gaussian negative log-likelihood on the available evaluations (x, z)' But then, equation 1 is not the NLL on z but on y."
>
> The NLL is minimized in $z$ and there is indeed no $y$ in equation 1.
>
> * "Why modeling the residuals this way and not the observations themselves? The split of modeling between a parametric and non-parametric part is not justified."
>
> This is in fact a key point of the paper. Modeling the residuals allows us to set a parametric prior that can transfer well across tasks by avoiding the numerical issues caused by different task and metric scales. Note that using a parametric prior also avoids the cubic complexity in the number of evaluations that makes it prohibitive to fit a GP on all task evaluations.
>
> * "Given that the authors are trying to aggregate information about the optimal hyperparameters from several tasks, they should not compare with single-task approaches, but with the simplest way to combine all the tasks. For instance: a) Normalize the outputs of every task. [...] b) Collect the z of all tasks and feed them into an existing GP black-box Bayesian optimizer. This is a very simple way to get "transfer learning" and it's unclear that the extra complexities of this paper (copulas, changes of variable with proper renormalization when the transformation is parameter free, etc) are buying much else."
>
> a) We asked ourselves the same question when designing our approach. For this reason a simple standardization to normalize tasks (where mean and std computed on each task separately) was evaluated in our ablation study in Section 5.2 and Table 3, where we probe the benefits of the Copulas and parametric priors against simpler alternatives. The results shows that standard normalizations are not competitive with our proposal, and these baselines perform poorly due to the scale issues met across different tasks and metrics. Given that this method was not seen in your first review, we renamed it and referenced it better in the ablation description to make it more visible.
>
> b) Following on the reviewer’s feedback, we ran additional experiments standardizing the outputs of every task and feeding these into warm-start GP. While this approach scales cubically in the number of observations, which makes it prohibitive with a large number of observations per task, our proposal of learning a prior does not have this limitation. Nonetheless, we applied this method by selecting the best hyperparameter evaluations from each related task for a total of 100 evaluations (so as to match the number of observations used in the warm-start GP baseline in the paper). The results show that our approach significantly outperforms this variant of warm-start GP, confirming as in Table 3 that standardization is not sufficient to transfer information successfully across heterogeneous tasks. We thank the reviewer for the suggestion and have incorporated this additional baseline into the paper.
>
> * "One of the main points of the variable changes is to normalize the scales of the different tasks. However, equations 1 adds together the samples of the different tasks (which, as pointed out by the authors might have different sizes)... the different dataset sizes will bias the solutions towards larger datasets... In fact, this will have the same effect as if the cost had different scales for different tasks, which is precisely the problem that the authors are trying to avoid."
>
> It is true that we do not normalize by the dataset size. Note that this is orthogonal to the scale issues we focus on: larger tasks will have larger gradient contributions but the scaling we propose still allows us to learn tied parameters across tasks as their scales are made similar. We thank the reviewer for the observation and revised the manuscript to discuss this option.
>
> * "Is d really the number of hidden nodes of the MLP? Or the number of outputs? Given that d is also the size of w, it seems it's actually the latter."
>
> We clarified in the main text that the number of outputs and nodes coincide.
>
> * "Explain why the EI approach is used for the second model (with the GP), but not for the first model."
>
> As the first model is stateless, the same hyperparameter minimizing EI would be sampled at each step (as opposed to the GP-based model). This is why we perform Thompson sampling instead.

---

### Official Review · AnonReviewer3 · 2019-10-27
**Official Blind Review #3**

**Rating:** 6

**Review:**

Summary:
This paper proposes a new Bayesian optimization (BO) based hyperparameter searching method that can transfer across different datasets and different metrics. The method is to build a regression model based on Gaussian Copula distribution, which maps from hyperparameter to metric quantiles. The paper shows that by leveraging this estimation using some specific (sampling) strategies, it is able to improve over other BO methods.

The high-level idea of this paper seems sound to me -- that improves standard BO to generalize across datasets and metrics by learning a mapping between the space of hyperparameters and metrics. While I am not an expert in this area, the derivation looks sound to me, and the evaluation results of this paper are comprehensive to show that CGP seems to outperform a number of methods.

**Experience Assessment:**

I do not know much about this area.

**Review Assessment: Checking Correctness Of Derivations And Theory:**

I did not assess the derivations or theory.

**Review Assessment: Checking Correctness Of Experiments:**

I assessed the sensibility of the experiments.

**Review Assessment: Thoroughness In Paper Reading:**

I made a quick assessment of this paper.

---

> ### Author Response · Authors · 2019-11-12
> **Response to Reviewer 3**
>
> We would like to thank you for the positive feedback.

---

### Author Response · Authors · 2019-11-12
**Response to reviewers**

We thank all the reviewers for their valuable feedback and uploaded a new version of our paper addressing their comments. With respect to Reviewer 2's comments on baselines, we would like to highlight that a baseline with standard normalization was present in the first version of our manuscript to demonstrate the benefits of the Copula transformation. Following on the feedback, we added the proposed variant of warm-start GP using standardization as well as the warm-start GP baseline suggested by Reviewer 1. Our approach outperforms both these additional baselines. In addition, we discussed the given references and performed the extra analysis proposed by Reviewer 1 to improve the understanding of task distributions.

---

### Decision · Program_Chairs · 2019-12-19

**Decision:**

Reject

**Comment:**

This paper tackles the problem of transferring learning between tasks when performing Bayesian hyperparameter optimization. In this setting, tasks can correspond to different datasets or different metrics. The proposed approach uses Gaussian copulas to synchronize the different scales of the considered tasks and uses Thompson Sampling from the resulting Gaussian Copula Process for selecting next hyperparameters.

The main weakness of the paper resides in the concerns raised about the experiments. First, the results are hard to interpret, leading to a misunderstanding of performances. Moreover, the considered baselines may not be adapted (they may be trivial). This might be due to a misunderstanding of the paper, which would align with the third major concern, that is the lack of clarity. These points could be addressed in a future version of the work, but it would need to be reviewed again and therefore would be too late for the current camera-ready.

Hence, I recommend rejecting this paper.